

# Extreme sensitivity of the northeastern Gulf of Lion (western Mediterranean) to subsurface heatwaves: Physical processes and devastating impacts on ecosystems in the summer of 2022

Claude Estournel[1], Tristan Estaque[2], Caroline Ulses[1], Quentin-Boris Barral[1], Patrick Marsaleix[1]

[1]Université de Toulouse, LEGOS (CNES/CNRS/IRD/UT3), Toulouse, France
[2]Septentrion Environnement, Marseille, France

*Correspondence to*: Claude Estournel (claude.estournel@cnrs.fr)

**Abstract.** In the summer of 2022, an atmospheric situation characterized by persistent anticyclonic anomaly caused an extreme marine heatwave in the western Mediterranean. Time series of temperature profiles at various points along the

northeastern coast of the Gulf of Lion (NW Mediterranean) showed exceptional temperatures down to depths of 30 m and massive mortality of benthic species. A hydrodynamic numerical simulation was used to analyze the physical processes responsible for this subsurface heatwave in a region where the climatology in summer is characterized by northerly winds inducing upwelling alternating with low winds. Firstly, the recurrence of heatwaves limited to the surface is demonstrated, triggered when upwelling stopped and warm water from the Northern Current intruded on the shelf. More importantly, in

August and early September 2022, two episodes of southerly and easterly winds of 8 to 10 m·s⁻¹ occurred. The oceanic response to these winds was an alongshore cyclonic current advecting warm water onto the shelf and a downwelling of this warm water to depths of the order of 30 to 40 m. A large part of the Gulf of Lion coast was warmed by these events. However, the northeastern part of the shelf, on either side of the city of Marseille, was by far the area most affected in depth, due to the combination of the proximity of the warm surface waters of the Ligurian coast advected by wind-induced currents

and the local acceleration of the wind by the continental topography, which intensifies the downwelling of these surface waters. These events are rare in summer, but their impact on the rich benthic ecosystems that characterize the region is dramatic, and will only increase with the warming trend in surface waters, already close to 1 °C for the last decade.

## 1 Introduction

With global warming, the number, intensity and duration of marine heatwaves (MHW) are on the rise, and MHW events

occur throughout the year (Fox-Kemper at al., 2021). Impacts on marine organisms are likely to be most marked in coastal areas. Benthic communities, being sedentary or having only limited mobility, are indeed particularly vulnerable to extreme temperatures (Lejeusne et al., 2010; Hughes et al., 2017; Garrabou et al., 2022) rather than to the exceedance of climatological values likely to occur throughout the year. This is the case for coral, for example, with a bleaching threshold defined for the Hawaiian Islands when surface temperature exceeds the maximum monthly mean by 1 degree (Glynn and



D'Croz, 1990). When biological impacts are considered, it is therefore summer heatwaves that should be particularly monitored.

While surface MHW have been documented over several decades from satellite SSTs (Sea Surface Temperatures), in-situ data documenting temperatures in the first few tens of meters below the surface are infrequent, and their spatial
representativeness varies greatly from site to site, particularly in coastal areas which are subject to unique dynamical processes due to the shallow depths and proximity to land (Schaeffer et al., 2023): on the one hand, irregularities in the coastline and submarine topography condition horizontal alongshore and cross-shore exchanges; on the other, spatial variations in the wind (potentially linked to continental relief) can generate localized coastal upwelling/downwelling depending on the orientation of the coastline relative to the wind direction. In contrast to upwelling, which counteracts the
occurrence and intensity of heatwaves in the first few tens of meters of the water column (defined here as the sub-surface) and is identifiable on the SST, downwelling favors the penetration of warm surface water at depth, possibly beneath the stratified surface layer, and is generally not identifiable from the SST, especially as surface water can be cooled by air-sea heat fluxes (Schaeffer et al., 2023). Tracking coastal subsurface heatwaves is therefore impossible using surface information alone. As underlined by Schaeffer et al (2023), the best proxy of sub-surface temperature extremes appears to be wind
anomalies since sub-surface MHW events and years characterized by many sub-surface MHW days are predominantly associated with wind-driven downwelling. Numerical models are the only way to document them at high spatial (horizontal and vertical) and temporal resolution. However, their accuracy is a prerequisite when it comes to exceeding a tolerance threshold.

Since 1982, the Mediterranean SST has been warming at a mean rate of ∼0.35 °C/decade (Pastor et al., 2020), compared with the global mean increase of 0.15 °C/decade (Fox-Kemper et al., 2021). For the 2010-2019 decade, Garrabou et al. (2022) showed an increase in warming for an average of seven coastal sites in the north-western Mediterranean, with values for the decade of 0.9 °C at 5 m and 0.6 °C at 35-40 m.  Superimposed on this long-term warming, several authors have noted increased frequency, intensity and duration of MHW over the last four decades (Simon et al., 2022; Juza et al., 2022; Pastor
et al., 2023) and an acceleration in recent years. Recently an exceptionally long-lasting and intense MHW, started in May 2022 (Martinez et al., 2023) and persisted until spring 2023 (Marullo et al., 2023). According to the latter authors, the intensity of this MHW was comparable to that of the 2003 event, which is the most intense case ever occurred in the last decades. The 2022 MHW was attributed to an atmospheric heatwave in Western Europe due to a persistent anticyclonic anomaly (Guinaldo et al., 2023) exacerbated by climate change (Faranda et al., 2023). Never-before-recorded temperatures
were observed at the surface of the western Mediterranean during this summer (Guinaldo et al., 2023). In the Marseille area, east of the Gulf of Lion, temperatures exceeded 26 °C at 20 m for 4 days in August and 25 °C down to 30 m for 3 days in early September (see more details in Grenier et al., 2022; Estaque et al., 2023). Following these last authors, depths from 25-



30 m were exposed for the very first time to temperatures above 25 °C, considered as a potentially lethal acute heat stress threshold (Crisci et al., 2011).

65

During the heatwaves of 1999 and 2003, large scale (>1000 km coastline) mass mortality events have affected numerous species of benthic invertebrates in the northwestern Mediterranean (Bensoussan et al., 2010 and references herein) indicating that they are living near their upper thermal thresholds. In the Parc National des Calanques south-east of Marseille, high temperatures in the summer of 2022 had an unprecedented impact on the mortality of numerous species, especially for the red gorgonian, *Paramuricea clavata*, and the red coral, *Corallium rubrum*, affecting depths down to 30 m (Estaque et al., 2023). Other emblematic species, such as sponges, have totally disappeared around Marseille at depths of down to 25 m, with only a few individuals surviving between 25 and 30 m (Grenier et al., 2022). The first sponge mortalities were recorded when temperatures exceeded 26 °C.

75 The aim of this article is to understand the interweaving of physical mechanisms from the large to the local scale at the origin of the summer 2022 heatwave, extreme in its intensity and impacts in the region of Marseille. To achieve this, we use numerical modeling checked against a dense network for top 40 m temperature measurement, called TMED-Net, which documents surface and subsurface heatwaves.

## 2 Main characteristics of the study site

The Gulf of Lion (Fig. 1) comprises a broad, crescent-shaped continental shelf enclosed between two straight coasts, the Ligurian coast to the northeast and the Catalan coast to the southwest, bordered by a narrow shelf and a steep continental slope, along which the Northern Current transports from northeast to southwest warm and low saline waters originating from the south of the basin and ultimately from the Atlantic.





Figure 1: Location and topography of the study area (background map from © Google Earth). The prevailing Mistral and Tramontane winds, the Northern Current, regional branch of the general circulation are indicated as well as the different places named in the text. The green solid circles indicate the TMED-Net observation points, Méjean, Banyuls-sur-Mer and Villefranche-sur-Mer on which comparisons with the model are focused, and Riou and Cap Sicié. The insert is a more general map of the western Mediterranean, in which the area is indicated by a white rectangle.

The north-western Mediterranean is a mosaic of contrasting hydrological situations due to bathymetric constraints and complex wind regimes. In the Gulf of Lion, the prevailing winds blow from land (north to northwest, referred to in the following as the northerly wind for simplicity's sake) and are channeled by orography, the Rhone valley to the north, where the Mistral blows, and the passage between the Pyrenees and the Massif Central to the west, where the Tramontane blows. As a result, winds are much stronger in the Gulf of Lion than along the Catalan and Ligurian coasts. These northerly winds produce discontinuous coastal upwelling to the north of the zone, from around 3.5°E to 5.8°E (Millot, 1990), leading to surface cooling in summer that can exceed 10 °C in 1 to 2 days (Odic et al., 2022). Using criteria not detailed here, based on the temperature anomaly with respect to climatology and the cooling between two successive days, the frequency of these events near Marseille between 2012 and 2022 averages 6.3 per summer (Barral pers. comm.). For the same winds (in this

case, Tramontane), the west coast is on the contrary favorable to downwelling (Bensoussan et al., 2010; Odic et al., 2022),
producing less variable surface temperatures but nevertheless presenting lower maximum values than to the east (Pairaud et
al., 2014) due to the stronger and more frequent winds (Obermann-Hellhund et al., 2018) and a reduced influence of the
warm Northern Current.

The second type of wind blows across the western Mediterranean from southwest to southeast, but locally these winds are
strongly influenced by the relief of the islands and the mainland, which can strongly modify their intensity and direction.
They generally enter the Gulf of Lion from a direction that varies from east after following the Ligurian coast, to southeast as
they bypass Corsica and Sardinia, and to south. In what follows, we will refer to these winds as southeasterly winds in
accordance with several authors (*e.g.* Millot, 1990; Odic et al., 2022). Rare in July-August, their frequency and intensity then
increase sharply to peak in October-November (Odic et al., 2022), when it is frequently accompanied by precipitation on the
mainland, which can lead to flash flooding of rivers, possibly accompanied by significant material and human damage
(Ducrocq et al., 2014; Drobinski et al., 2014). These winds, which can reach 25 m·s$^{-1}$ in winter, induce cyclonic circulation
around the Gulf of Lion, with currents of several tens of cm·s$^{-1}$, accompanied by intensified downwelling to its southwestern
tip due to the acceleration of currents linked to the narrowing of the continental shelf (Ulses et al., 2008; Mikolajczak et al.,
2020).


The region around Marseille includes to the east the "Parc National des Calanques", characterized by remarkable marine
habitats (e.g. *Posidonia* meadows, coralligenous reefs, semi-dark caves, submarine canyons), 14 of which are considered
rare and fragile (https://www.calanques-parcnational.fr/en/marine-habitats) and which presents a high risk of mass mortality
associated to thermal stress for the red gorgonian (Pairaud et al., 2014) as well as for other cnidarians, numerous sponge
species, bryozoans and tunicates. To the west of Marseille, the "Parc Marin de la Côte Bleue" is another marine protected
area, also hosting a great marine biodiversity similar to that found in the "Parc National des Calanques", but at relatively
shallower depths. In this last area, the presence of remarkable mesophotic 'giant' *Paramuricea clavata* forests is particularly
noticeable (Sartoretto et al., 2023).

## 3 Material and Methods

We use a numerical simulation of the entire Mediterranean basin similar to that described in Estournel et al. (2021). The
simulation is based on the 3D primitive equations SYMPHONIE model described in Marsaleix et al. (2008, 2006) and
Damien et al. (2017), with turbulence closure and convection parameterization detailed in Estournel et al. (2016). The VQS
(vanishing quasi-sigma) vertical coordinate (Estournel et al., 2021) is used with 60 levels. The horizontal resolution in the
area of interest is around 1900 m, which may seem coarse for an application very close to the coast, but proved sufficient to
achieve our objective. The model is initialized and forced in the Gulf of Cadiz (Atlantic Ocean) from operational oceanic





analysis produced by MERCATOR OCEAN International and at the surface by hourly forecasts of the ECMWF operational meteorological model through COARE bulk formulas for the turbulent air/sea fluxes. The simulation was initialized in May 2011. Comparisons with monthly satellite SST taken at two seasons, 6 and 7 years after model initialization, show a very good representation of large-scale features present in the whole basin but also of various smaller structures (Estournel et al., 135  2021).

At the coastal scale, the simulation is compared with TMED-Net (https://t-mednet.org/) observations taken at an hourly frequency every 5 m between 5 m and 40 m with autonomous sensors fixed to the seabed rocky substrate. The summer of 2022 was marked around Marseille by mass mortality of sponges and gorgonians (see introduction). This region includes 140  various TMED-Net observation points stretching from the Côte Bleue to Cap Sicié. For our study, we focused on the Méjean site located 10 km west of Marseille (Fig. 1), where subsurface warming was generally strongest. We also took a broader view of the north-western Mediterranean, in order to visualize the specificity of the Marseille region. To do this, we compared temperature trends at Méjean with those recorded at TMED-Net points in Banyuls-sur-Mer and Villefranche-sur-Mer, located around 200 km to the south-west and north-east of Marseille (locations on Fig. 1). As mentioned in section 2, 145  Banyuls-sur-Mer on the west coast of the Gulf of Lion is subject to frequent downwellings, and Villefranche-sur-Mer on the Ligurian coast is strongly impacted by the Northern Current.

To link the intensity of the 2022 MHW with the extreme impacts observed on benthic communities to depths down to 30 m, we used a compiled dataset acquired by scientists and marine protected area managers in 2022, after the MHW. This dataset 150  is the largest available for monitoring a mass mortality event for benthic organisms in the Mediterranean. The data was acquired using the methods detailed by Estaque et al. (2023), who analyzed some of the data in detail for the "Parc National des Calanques" area. Here, the dataset consists of data on the health of 18,465 colonies of red gorgonian (*P. clavata*), white gorgonian (*Eunicella singularis*) and yellow gorgonian (*Eunicella cavolini*), from 298 populations dwelling between the surface and 40 m depth. The data used allow us to compare the impact between the "Parc National des Calanques" (PnCal), 155  the "Parc Marin de la Côte Bleue" (PMCB), the "Cap Sicié" (CS), and the "Parc National de Port-Cros" (PNPC), in relation to the differences in intensity of the 2022 MHW event.

## 4 Results

### 4.1 Assessment of the simulation and hydrological characteristics during summer 2022

Fig. A1 (Appendix A) shows the temporal evolution of measured and simulated temperature at Méjean between 5 and 40 m 160  and between 2012 and 2022. The simulation shows a cold bias of around 0.5°C at all levels, with uneven performance from year to year, particularly at 20 m depth, suggesting an uncertainty in the representation of the thermal gradient of the highly stratified layer beneath the surface mixed layer. The correlations between the observed and simulated series are above 0.95.





Rapid temperature variations in summer, reflecting the succession of upwelling and stratification events, are visible at all levels and are well synchronized, indirectly indicating a correct representation of wind in the simulation. The maximum of the series at 5 m is 27.8 °C in hourly values, reached in August 2022 (28.5 °C at Riou 20 km further south, sea location on Fig. 1). The duration of exceedance of the 25 °C value in 2022 is also the longest of the decade. The simulation agrees with observations on these two extremes of temperature and duration.

Figures 2a to 2d zoom in on Fig. A1 from June to the end of September 2022. Correlations between observed and simulated series range from 0.88 at 40 m to 0.96 at 5 m.  The model bias is maximum at 5 m (-1 °C) and varies between -0.18 and 0.15 °C at deeper levels. Warming episodes, highlighted in red, correspond to periods of continuous warming of at least 5 °C either at the surface or at 20 m. Northerly winds inducing cooling episodes are highlighted in blue when they reach 10 m·s$^{-1}$.

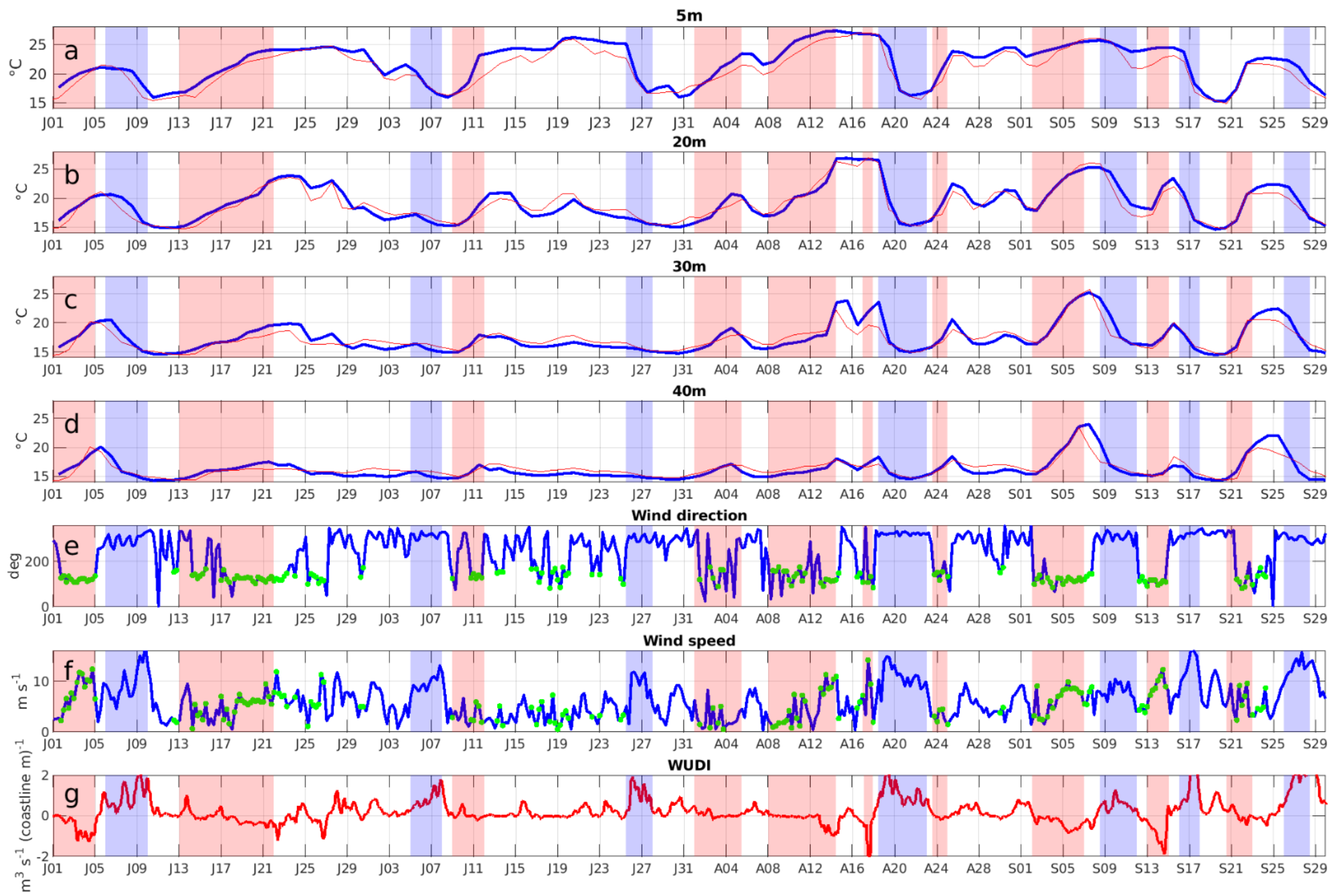

**Figure 2: Observed (blue) and simulated (red) daily averaged temperature time series at the Méjean point of the TMED-Net network during the summer of 2022 (June to September), from 5 m to 40 m, as indicated above the figures a-d. Wind direction and intensity (6-hour moving average), near Marseille, given by the ECMWF operational model (e and f). The southeasterly winds are indicated in green. Wind-induced Upwelling and Downwelling Index (6-hour moving average) calculated at Méjean (g). Warm events are highlighted by red stripes and northerly winds associated with cooling by blue stripes.**



Events above 25 °C at 5 m followed one another from July 19 to September 9, with contrasting signatures further down. The
event that peaked at the surface on July 19 resulted in an increase of around 3 °C at 20 m, but was much less pronounced at
greater depths. In contrast, the mid-August event was almost as warm at 20 m as at 5 m, with a signature at 30 m around +5
°C on August 14, shortly after the surface maximum. The early September event was almost uniform down to 30 m (T>25
°C) and reached 24 °C at 40 m.

Figure 3 compares the local characteristics of the subsurface heatwave observed by the TMED-Net network and simulated
along the anticlockwise pathway of the Northern Current  at Villefranche-sur-Mer, Méjean, Banyuls-sur-Mer. In contrast to
Méjean, Villefranche-sur-Mer's summer temperatures in 2022 did not vary greatly. The warm period extends over more than
two months and generally involves a well-stratified surface layer. In Banyuls-sur-Mer, surface water temperatures are also
more continuous than in Méjean, but significantly lower than in Villefranche. On the other hand, the heat is distributed over
a much thicker layer, evoking the recurrent presence of downwellings. The simulation reproduces the thermal regime at the
various sites with shortcomings such as the underestimation of the surface layer temperature at Villefranche-sur-Mer.

### 4.2 Meteorological characteristics during summer 2022

Figures 2 e,f show the wind direction and intensity of ECMWF meteorological model near Marseille. For greater visual
clarity, hourly winds are averaged over 6 hours. Figure 2g illustrates the wind-induced upwelling and downwelling index
(WUDI), as outlined by Odic et al. (2022), calculated with the ECMWF at the Méjean probe location. This index quantifies
the horizontal Ekman transport within the model, exhibiting a positive (negative) value, indicative of upwelling
(downwelling). In line with climatology, the strongest winds blow from the mainland. Whenever the north, northwest wind
exceeds 10 m·s$^{-1}$, a surface temperature drop of at least 5 °C is observed and simulated over the following days. This is the
case for events starting on June 6, July 5 and 26, August 18, September 9, 16 and 26. These upwellings are also visible
below the surface (especially if warming has taken place beforehand, as in the cases of June 6 and September 9). In this
respect, it's worth noting that temperatures drop at depth around 24 hours before at the surface (see for example the June 6
event). In all of the aforementioned instances, the WUDI is greater than +0.5 m³·s$^{-1}$·(coastline m)$^{-1}$, which is consistent with
the previous identifications of upwelling.

During summer 2022, warm events are characterized by lower wind intensities than northerly wind events. They are
classified here as either weak winds (typically below 5 m·s$^{-1}$), or moderate southeasterly winds (typically above 5 m·s$^{-1}$ and
rarely above 10 m·s$^{-1}$). All cases of weak or moderate southeasterly winds are indicated in green in Fig. 2 e,f and are all
located in warm periods. Specifically, the periods in question are June 1-5, 13-22, July 9-12, August 1-4, 8-14, 23-25, and
September 2-7, 13-15. For moderate wind speeds, the negative WUDI index below -0.5 m³·s$^{-1}$·(coastline m)$^{-1}$ suggests a
potential contribution of downwellings to warming episodes.





**Figure 3: Hovmöller diagrams of the temperature (°C) profiles (m) given by the TMED-Net probes (a, c, e) and the model (b, d, f) at the locations of Villefranche-sur-Mer (a, b), Méjean (c, d) and Banyuls-sur-Mer (e, f) (see locations on Fig. 1). Isotherms every 2.5°C. The probe diagrams are smoothed vertically and over 24 hours in order to facilitate optimal viewing.**





In order to characterize southeasterly winds over the summer of 2022 in relation to climatology, statistics have been compiled over the period 2012-2022. Each month, the time integral of negative WUDI (i.e. downwelling index) is calculated. Figure A2 (Appendix A) shows the climatology of this index, the dispersion over 11 years and the value for 2022. Summer, and especially July and August, show very low values compared with autumn and winter, which also show very high interannual variability. For the summer of 2022, downwelling dynamics is particularly weak in July, while in

August it is the strongest in the series, while remaining low compared to the rest of the year.

Figure 4 shows the surface wind fields during the two warmest events at depth (30 m), on August 13-14 and September 5-7. In both cases, a low-pressure area was present over western Europe (western France in the first case, and the British Isles in the second) leading to a southerly flow over the Mediterranean, which penetrated the mainland in the Gulf of Lion. This flow

in the lower layers of the atmosphere was strongly influenced and accelerated by the relief especially by Corsica and Sardinia, which were bypassed by veins of wind between and around the two islands. These wind veins converged in the eastern Gulf of Lion, here influenced by the topography of the Provençal coast and the edge of the Alpine chain, producing a noticeable acceleration in the vicinity of Marseille.

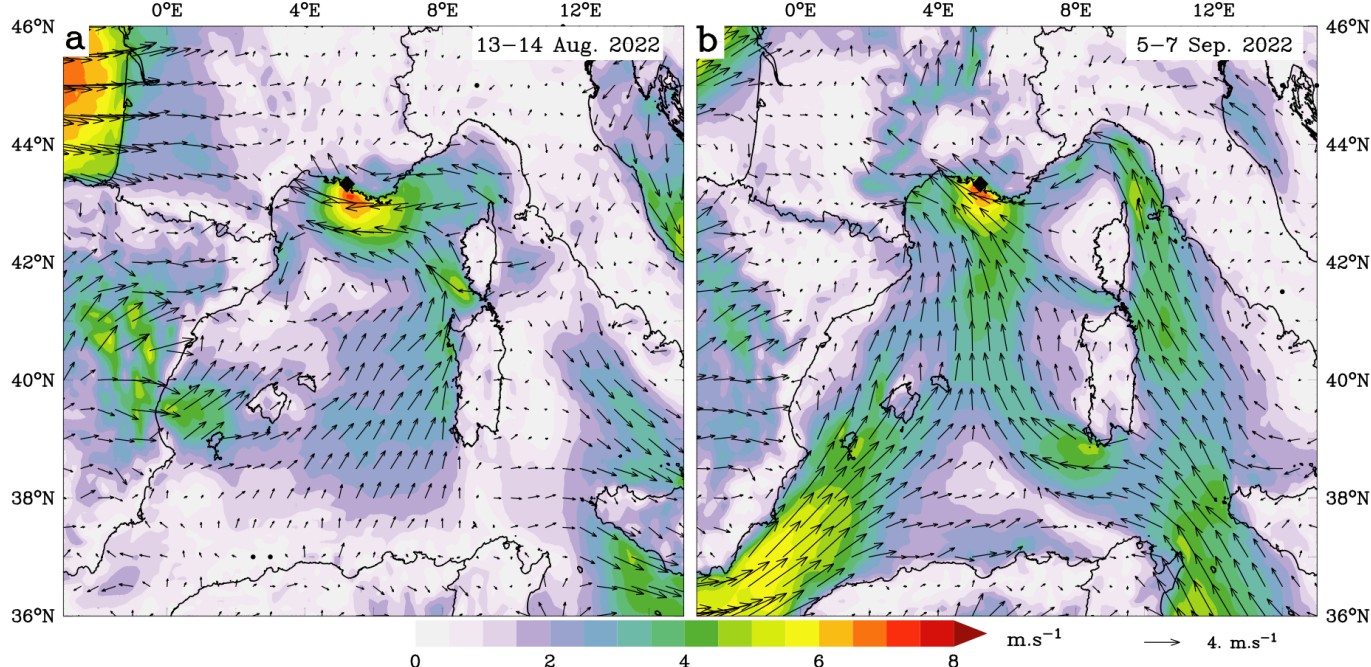

**Figure 4: Wind field (m·s⁻¹) at 10 m averaged over the two warmest subsurface periods: (a) 13-14 August 2022 ; (b) 5-7 September 2022. The black diamond stands for the Méjean observation point.**

### 4.3 Marine heatwaves

The warmest event at the surface and at 20 m, that triggered sponge mortality, occurred in mid-August. At 30 and 40 m, temperatures peaked later, during the September 6 event. Events were longest at the surface : the major event in mid-August



lasted around 2 weeks. While warm events were characterized by weak wind or moderate southeasterly wind conditions, they were stopped by northerly wind gusts (Fig. 2). The points discussed in the following 3 subsections are warming during weak wind conditions (4.4.1) and during moderate southeasterly winds (4.4.2), and sensitivity of sub-surface warming to the duration of southeasterly winds (4.4.3). From mid-August onwards, impacts appeared as massive mortalities down to 25 m, and more moderately at 30 m. A sensitivity study will explore the temperature response to an extension of the wind period.

**4.3.1 Case of weak winds: Impact of pre-existing upwelling**

Five warming episodes were considered: June 13-18, July 9-12, August 1-4, August 8-12 and August 23-25, characterized by winds of less than 5 m·s⁻¹, mostly blowing from the southeast, with occasional short spells from the northwest (Fig. 2). For each of these events, temperature at 5 m increased by 4 to 7 °C (on average 5.4 °C). At 20 m, the warming was slightly lower (averaging 5 °C), then decreased at 30 m to an average value of 3 °C and at 40 m to 1.9 °C (more details in Table A1

in Appendix A). While the warming at depth could be attributed to downwelling, this cannot be the case at 5 m, as this would imply a considerable temperature gradient between the surface and 5 m. Let's now consider the case of the July 9-12 event, for which the warming at 5 m was 6.7 °C in two days and 5.4 °C at 20 m. Net solar radiation at the surface averaged over 24 hours was 320 W·m⁻². If we consider that this flux was absorbed by a 20 m thick layer, the warming resulting from this absorption over 2 days would be 0.8 °C, i.e. much lower than observed, which rules out the hypothesis of warming by

the atmosphere (the solar flux in this case). For each event considered, the warming at Méjean started between 1 and 3 days after the end of a northerly gale (varying in intensity from 8 to 15 m·s⁻¹) and of the associated upwelling. In such conditions of northerly wind cessation during stratified periods, Barrier et al. (2016) have highlighted intrusions of the Northern Current on the Gulf of Lion shelf with a delay of 1 day after the wind relaxation. The processes associated with upwelling relaxation explain the observed and simulated surface warming, which, in our case, implies that the water mass advected by the

Northern Current intrusion was much warmer than that present on the Gulf of Lion prior to the upwelling. The eastern part of the Gulf of Lion is particularly favorable to this situation, as it borders to the east of 6°E, an area much less exposed to northerly winds and therefore much warmer at the surface.

As an example, Fig. 5 a,b,c show the surface temperature and current on July 8, 12 and 15: at the peak of the northerly wind

and upwelling on July 8, during and at the end of the warming phase on July 12 and 15. The intrusion of a branch of the Northern Current producing the renewal of surface water by the advection of warm water is visible along the coast in the Marseille area after the stop of the northerly wind. The advected water mass was located east of 6°E on July 8 (Fig. 5a), then 50 km further west on July 12 (Fig. 5b) and around 100 km (up to 5°E) on July 15. (Fig. 5c). It is interesting to note that the strong, sustained northerly wind (~ 10 m·s⁻¹ for 4 days) which preceded the warming episode, produced marked hydrological

structures (Fig. 5a) such as the upwelling front stretching southwards off Marseille and, further west, organized cross-shore currents. These structures, which persist for a few days, seem to hinder the progress of warm waters westwards.



### 4.3.2 Case of moderate to strong southeasterly winds

We consider here the two cases associated with the warmest temperatures at 20 m: the second part of the major August event, from August 12 to 14 extended by a new peak on the 17th, and the period from September 2 to 7. In both cases, the

southeasterly wind whose average is presented in Fig. 4, exceeded 8 m·s$^{-1}$ and the WUDI index fell, showing coastward transports of over 0.5 m³·s$^{-1}$·(coastline m)$^{-1}$ that lasted 2 days (Fig. 2). In contrast to the low-wind cases, warming was minimal at the surface, as temperatures were already high at the start of both events. In terms of the vertical extension of warming, the difference with low-wind cases is striking (Fig. 2, Table A1 & Fig. 3). Warming increased down to 30 m, with values of 5.5 °C in August and 8.8 °C in September (compared with 2-3.6 °C for weak winds). For the first event, for which

the southeasterly wind is of shorter duration, warming is 2 °C at 40 m, and 8.5 °C for the second. Again in contrast to the low-wind cases, the northerly wind preceding the September 6 event was weak (less than 10 m·s$^{-1}$) and of short duration (from August 31 to September 1). As a result, the associated upwelling (Fig. 5d) and the currents on the inner shelf were poorly developed, as can be seen from the comparison with Fig. 5a. We cannot therefore expect a massive intrusion of the Northern Current and significant warming associated with the cessation of upwelling. In this case, the warming was due to

the southeasterly wind causing surface water to pile up against the coast, inducing the downwelling of warm surface water, and finally, through geostrophy, the development of an alongshore westward jet. The persistence of the easterly wind (6 days without interruption) and the weak currents that prevailed before the easterly wind, favored the establishment of this circulation at the whole continental shelf scale, warming a large part of the coastal zone of the Gulf of Lion between September 1 and 6, as shown in Fig. 5f. At a depth of 30 m, Fig. 6a-c show the pre-event and peak temperatures and their

differences, indicating that the eastern Gulf of Lion was considerably warmed up by around 7°C over 600 km$^2$ from 5.62°E east of Cassis to 4.85°E near the Rhône mouth. Further west, the coastal strip extending up to the 60 m isobath was heated by 2 to 4 °C. The similarity between the area of maximum warming and that of maximum downwelling favorable winds (Fig. 4b) is remarkable.





**Figure 5: Surface temperature (°C) and current simulated for the surface warming event of July (a,b,c) : (a) July 8 (during the upwelling), (b) July 12 (the day after the wind stopped), (c) July 15. Same for the subsurface warming event of September (d,e,f): (d) September 1, (e) September 4, (f) September 6. The green point stands for the Méjean TMED-Net observation point. The green dashed lines are the 200 and 1000 m isobaths.**





**Figure 6: (a-c): September event. Temperature (°C) at 30 m depth before the September subsurface warming event (a : September 1) and at the peak of the event (b: September 7); (c) warming between these two dates; (d) August event. Temperature difference (°C) at 30 m depth on August 16 between the test (extension of the southeasterly wind period) and the reference simulations. The green dashed lines are the 200 and 1000 m isobaths.**

### 4.3.3 Sensitivity of heat penetration in depth to the duration of the southeasterly wind

We consider the mid-August event at Méjean, the warmest of the summer at the surface. We described it as a succession of two main events (Fig. 2e-f) : one with weak winds from August 8 to 12, and the second with sustained southeasterly winds from August 12 to 14 (Fig. 4a), when sponge mortality in the Bay of Marseille began to be observed. This mortality was massively concentrated in the 0-25 m layer and more sparsely at 30 m. The second event is not detailed because it is similar to the one in September, except that, as mentioned in the previous paragraph, the duration of the southeasterly wind was shorter and the deep warming was lower. To understand the relation between warming and the duration of the southeasterly wind, we conducted a sensitivity study. The sustained wind event lasted about 40 hours. Knowing that the occurrence of southeasterly winds is rare in summer and that August 2022 was well above average for this occurrence (Fig. A2), we limited ourselves, to remain realistic, to extend the southeasterly wind by one day. This modification was made during the





310 weakening wind period from 1 PM on August 14 to 1 PM on August 15, where we substituted the stronger wind from the period of 1 AM on August 13 to 1 AM on August 14. The temperature at 30 m of the reference simulation being underestimated from August 15 to 18 by 3 °C on average (Fig. 2), we will subsequently present differences between the test and the reference simulations rather than the biased warming for each simulation. Fig. 6d represents this temperature difference at 30 m on August 16. As expected, the lengthening of the wind period results in increased warming. The temperature difference at 30 m from August 15 to 17 exceeds 3.5 °C around Marseille. The daily average temperature 315 observed on August 14 and 15 was 23.8 °C. If we transpose the results of the sensitivity test to the observed series, we deduce that the temperature at 30 m would have reached the value at 20 m. This would likely have led to a deepening of the mass mortality zone to 30 m instead of 25 m, and partial mortality at 35 m due to an increased warming of 1.3 °C compared to the reference simulation. It can be seen that the maximum warming along the Gulf of Lion coast is in the same place as in the early September situation (Fig. 6c). This is also where the southeasterly wind is strongest during the warming period 320 (Fig. 4a).

### 4.4 Impact on marine benthic communities

Under the influence of southeasterly wind regimes, which induced warm water downwelling to a depth of 30 m, and MHWs that exhibited heightened intensity across a 600 km² area between 5.62°E (east of Cassis) and 4.85°E (near the Rhône River mouth), gorgonian mortality was observed to be both more severe and extended to greater depths within this region. 325 Generally speaking, the impact on gorgonian populations appears to be most severe in the "Cap Sicié", "Parc National des Calanques", and "Parc Marin de la Côte Bleue" areas, particularly for the red gorgonian (*P. clavata*). For this species, which is the most sensitive, more than 50% of the populations monitored in the 20 to 30 m depth range in the PnCal showed a severe impact (> 60% of affected colonies), compared with less than 10% in Port Cros for the same depth range (Fig. 7). More specifically, in line with the MHWs observed, no impact has been recorded for the *E. cavolini* in the PNPC area, while 330 a sometimes severe impact has been recorded for this species in the PnCal and PMCB areas. The same observation was made for *E. singularis*, whose populations were low affected (10-30% affected colonies) in the PNPC area for the 10-20 m depth range and were not affected deeper, while the populations were sometimes severely affected (> 60% of affected colonies) down to 20 m in the PnCal area, and moderately affected (30-60% of affected colonies) in the PMCB down to 30 m. These observations suggest a greater impact at greater depths (down to 30m) for the gorgonian populations around the 335 Marseille region than within the PNPC.





**Figure 7: Map showing the severity of the impact of the mass mortality event on the gorgonian populations of the "Parc Marin de la Côte Bleue" (PMCB), the "Parc National des Calanques" (PnCal), the "Cap Sicié" (CS), and the "Parc National de Port-Cros" (PNPC). The impact on populations is represented by 10 m depth ranges between the surface and 40 m. See Estaque et al. (2023)**
**for more details on the monitoring method. Data for P. clavata in the PnCal are presented in more detail in Estaque et al. (2023).**

### 4.5 Variability of deep heatwaves on a regional scale

Figure 3 illustrates the major spatial variability of the upper layers temperature chronology on a regional scale. The upwelling and downwelling temperature alternations discussed at Méjean are much reduced at the other two sites. Villefranche-sur-Mer is characterized by the presence of warm water throughout the summer in the surface layer, due to

warming by the Northern Current flowing close to the coast and weak winds. Surface temperatures are frequently as high as the maximum recorded at Méjean on August 15. The two deep heatwave events studied at Méjean are also present at Villefranche-sur-Mer, also indicating the influence of the easterly wind in this region, albeit attenuated compared with the Marseille area (Fig. 3), which reduces the depth impacted by downwelling by about 10 m. At Banyuls-sur-Mer, the situation is different from the two other sites, with repeated temperature peaks almost reaching 40 m. The deep heatwave events of

mid-August and early September studied at Mejean are also present, albeit a few days late but temperature peaks are lower





than at the two other sites , reflecting the reduced influence of the Northern Current over most of the Gulf of Lion shelf and the frequent presence of dry northerly winds cooling the surface.

## 5. Discussion and perspectives

During the summer of 2022, the atmospheric heatwaves that hit Western Europe gave rise to extreme marine heatwaves
across the western Mediterranean. Although being a coastal zone relatively tempered by the recurrence of northerly winds and associated upwelling, the east of the Gulf of Lion and the very rich benthic ecosystems that it shelters have been exposed to short periods of exceptional temperatures.

The succession of hot and cold events in summer 2022 has been successfully reproduced by the numerical simulation: all
events have been simulated and the correlations are generally above 0.9. To go further, the precise reproduction of warm events is a major issue when it comes to determining the crossing of thermal thresholds representing the survival of marine species. In order to improve this precision, we identify two sensitive points here. On the one hand, a higher resolution of the horizontal grid will improve the description of the strong bathymetric gradients characteristic of the coastal area around Marseille and further east, along the Ligurian coast, and consequently the representation of horizontal and vertical
movements. On the other hand, the accuracy of the near-shore wind field is likely crucial for the accuracy of simulated temperatures due to the influence of topography on wind channeling and acceleration. Exploring the uncertainties of these two origins would be informative to improve the description of risk areas and to be able to predict extreme temperatures with better precision without unnecessarily increasing computational costs.

Despite the uncertainties, our modeling results provide insights onto the two types of heatwaves that were recorded in 2022. The first type of heatwave is a surface phenomenon. The physical processes at play are recurrent when the northerly winds and induced upwelling stop, and the wind speed following the northerly wind is less than $\sim 5$ m·s$^{-1}$. This succession of meteorological conditions causes an intrusion of warm water from the Northern Current along the northeast coast of the Gulf of Lion. In these situations, only the 5 m level exceeded 25 °C. In summary, these events renew the coastal surface water
mass by an advection of warm water until the following northerly gale which replaces the surface water with cold subsurface water. The second type of heatwave, more dramatic because it affected depths of 30 to 40 m, is generated by southeasterly winds (around 8 - 10 m·s$^{-1}$). Surface heating is still due to the advection of warm water from the east but it is pushed downward by wind-induced coastal downwelling. The strength and duration of the southeasterly wind are crucial parameters which determine the depth impacted.


On a regional scale ($\sim 200$ km), we have shown considerable variability in surface and deep heatwaves for two reasons: (i) the channeling of northerly and easterly winds by the continental relief, which produces localized upwelling and





downwelling, and (ii) warming by the Northern Current, whose influence is great along the Ligurian coast and reduced in the western part of the Gulf of Lion. The TMED-Net network is an extremely valuable tool for documenting this variability and

providing a database for validating high-resolution numerical models .

The severity of the 2022 summer for benthic species is the result of the superposition of two conditions, both exceptional, which are (1) the atmospheric heatwave which led to an exceptional warming of the surface of the western Mediterranean and (2) the two southeasterly wind events in mid-August and early September which caused these warm waters to plunge to

depth. These events are rare in summer as evidenced by the monthly downwelling index which is the highest in the decade for August, to which must be added the major event that occurs at the beginning of September. The occurrence of sustained southeasterly winds in summer, when surface temperatures are warmest, therefore constitutes a major danger for coastal ecosystems particularly for benthic species located above 40 m depth. Unfortunately, the coastal region around 30 km on either side of Marseille, with its remarkable habitats, is at far greater risk than the rest of the Gulf of Lion, due to the

acceleration of southeasterly winds caused by the topography, which locally intensifies downwelling. Indeed, we have verified that the acceleration shown here for the two major events of August and September exists in most of southeasterly wind situations occurring between June and September. This is indirectly confirmed by Odic et al (2022) who, using the wind from the ERA5 reanalysis to calculate upwelling and downwelling indices, showed that the Marseille vicinity is not only the most powerful upwelling zone in the northern Gulf of Lion, but also the most exposed to downwelling.


Global warming was proven to have contributed to the extreme temperatures experienced in the western Mediterranean during the summer of 2022. Exploring the coincidence of southeasterly winds and surface heatwaves in future climate scenarios would help anticipate the recurrence of massive mortality events and ultimately the disappearance of benthic populations, which are emblematic of the Calanques region of Marseille and of the Côte Bleue. Given the ongoing global

warming trends, an increase in the frequency and intensity of MHWs in the affected depth zones (0 - 30 m) and deeper is expected. These extreme thermal events are likely to severely impact the potential recovery of benthic organism populations, and the potential role of refuge from deeper populations is not certain (Bramanti et al., 2023). Certain benthic species, such as gorgonians, play a crucial role as ecosystem engineers, contributing significantly to habitat complexity (Verdura et al., 2019). The collapse of these species would therefore lead to a marked reduction in structural diversity, with cascading effects

on the broader ecosystem, including the disruption of ecological functions (Ponti et al., 2014) and the loss of vital ecosystem services (Estaque et al., 2023; Garrabou et al., 2021; Gómez-Gras et al., 2021). To mitigate these impacts, it is imperative to adopt multidisciplinary approaches that integrate ecological, oceanographic, and climatological data to better predict the occurrence and intensity of MHWs. Such strategies are essential for developing adaptive coastal area management and conservation efforts, with the goal of preserving the integrity of Mediterranean benthic communities and maintaining the

ecosystem services they provide. In a context where management plans are predominantly designed within a two-



dimensional framework (Jacquemond et al., 2024), this approach marks a critical advancement towards recognizing the ocean as a three-dimensional environment, particularly when establishing marine protected areas and conservation zones.

Finally, heatwaves will not only intensify in frequency and intensity with climate change but thermal stress will combine with other stresses whether linked to the long-term increase in anthropogenic CO2 or to the short-term impact of heatwaves on the chemical and biogeochemical composition of water. An extension of this study towards the impact of the 2022 heatwaves on oxygen and chlorophyll concentrations in the Gulf of Lion is planned.





**Appendix A**

| Surface event | 5 m heating | 20 m heating | 30 m heating | 40 m heating |
|---|---|---|---|---|
| June 13-18 | 4.8 | 4.2 | 2.3 | 1.8 |
| July 9-12 | 6.7 | 5.4 | 2.6 | 1.4 |
| August 1-4 | 4.1 | 4.7 | 3.6 | 2. |
| August 8-12 | 4.7 | 4.2 | 2. | 1.1 |
| August 23-25 | 6.8 | 6.4 | 4.8 | 3. |
| *Average* | *5.4* | *5.* | *3.* | *1.9* |
| **Subsurface event** | **5 m heating** | **20 m heating** | **30 m heating** | **40 m heating** |
| August 12-14 | 0.6 | 5.6 | 5.8 | 1.9 |
| September 2-7 | 2.1 | 7.5 | 8.8 | 8.5 |
| *Average* | *1.4* | *6.5* | *7.3* | *5.2* |


**Table A1: Identification of the different surface and subsurface warm events. Warming in °C observed at Méjean over each period at different depths (TMED-Net data).**





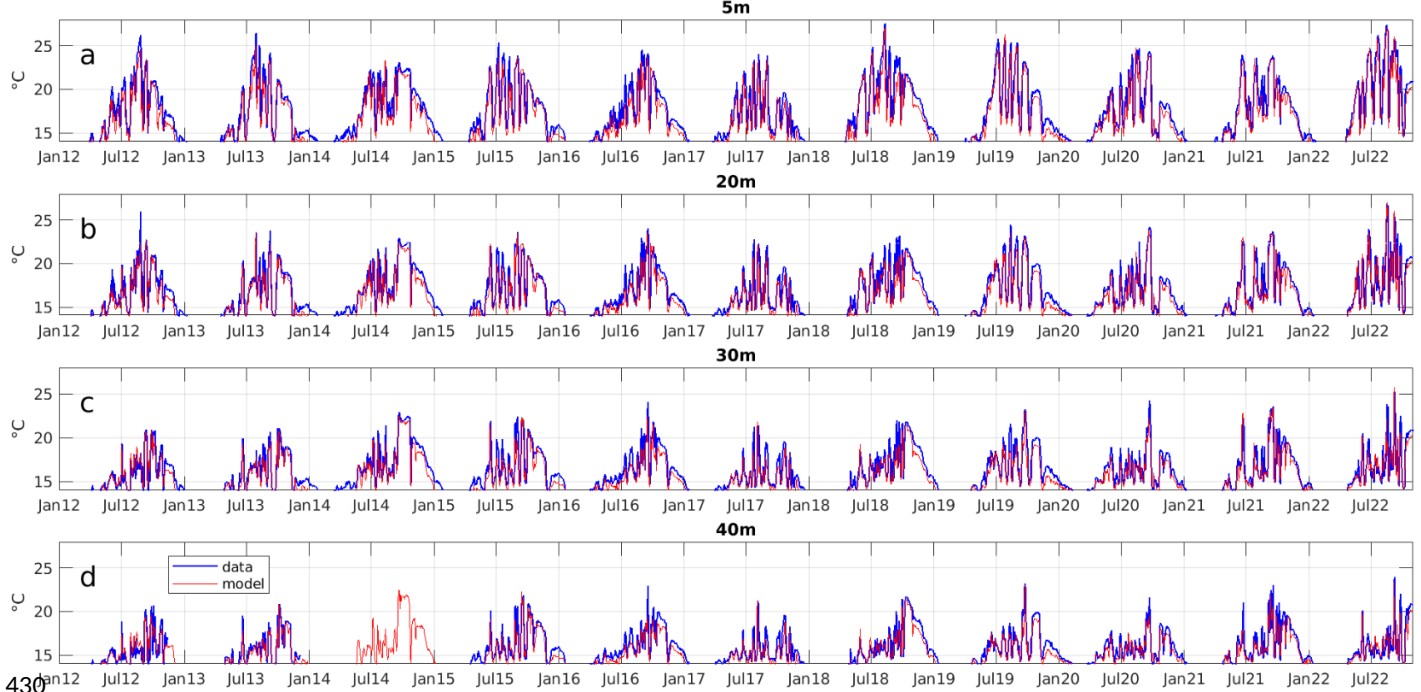


**Figure A1: Observed (blue) and simulated (red) temperature time series at the Méjean point of the TMED-Net network from 2012 to 2022 from 5 m to 40 m as indicated above the figures. The frequency shown is daily.**



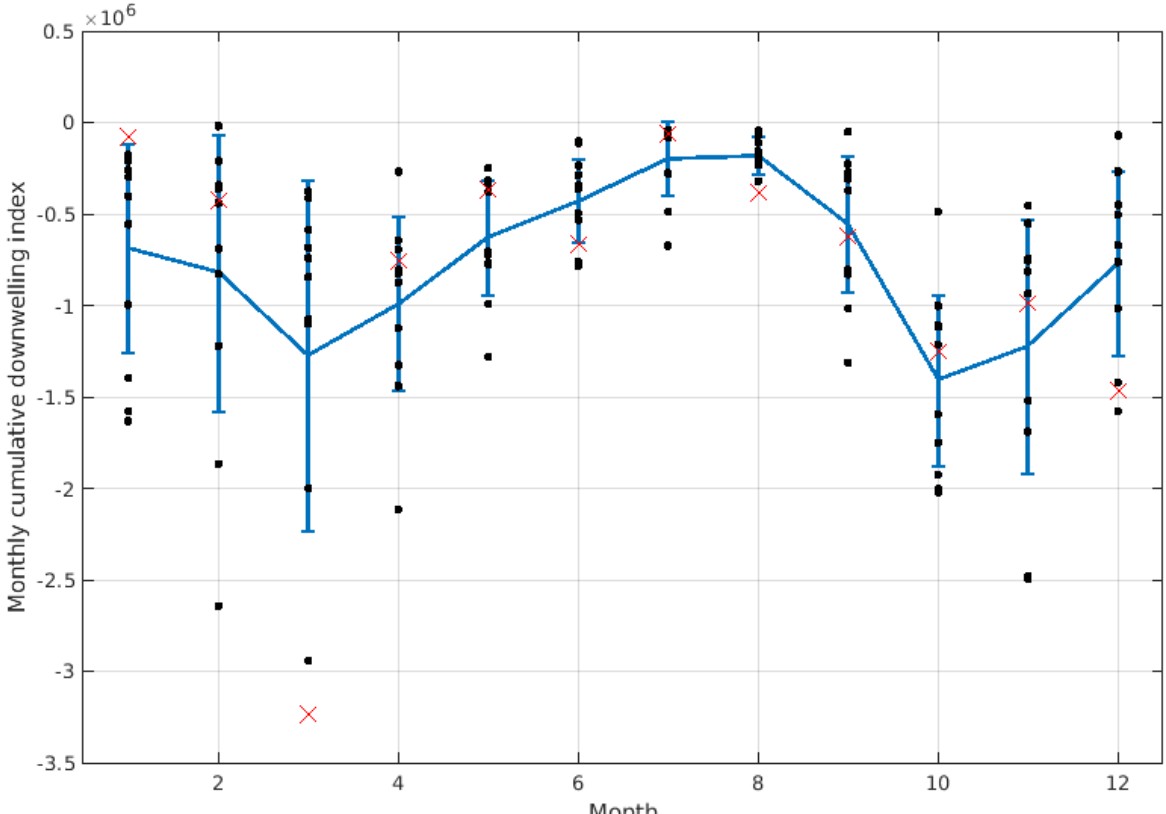

**Figure A2: Monthly cumulative downwelling index (m³·(coastline m)⁻¹ ) off Marseille. blue curve: climatology +/- standard**
**deviation calculated over 2012-2022. Black dots represent the monthly values for each year and the red cross is for 2022.**

**Author contribution**

CU and CE planned the study and conceptualized the paper. TE analysed the biological data.The model was developed by PM. CE and PM performed the simulations. QBB analysed the meteorological forcing. CE and TE wrote the first version of the paper. All co-authors edited and corrected the text.

**Competing interests**

The authors declare that they have no conflict of interest.

**Acknowledgments**

This study was funded by the OFB (Office français de la biodiversité) and ILICO littoral and coastal research infrastructure
(www.ir-ilico.fr) through the INTEGRATION workshop for the study of marine heatwaves along the French coastline. The



temperature data have been provided by the regional temperature observation network T-MEDNet, www.t-mednet.org, site "Méjean", Dorian Guillemain, OSU Institut Pythéas, site "Villefranche-sur-Mer", Nuria Teixido and Steeve Comeau, Laboratoire d'Océanographie de Villefranche-su-Mer, site "Banyuls-sur-Mer", Ronan Rivoal, Réserve Naturelle Marine de Cerbère Banyuls /Conseil Départemental des Pyrénées Orientales. For the health data on gorgonian populations, we would

like to thank Eric Charbonnel, Patrick Bonhomme, Pauline Vouriot, Stéphane Sartoretto, Quentin Schull, Bastien Mérigot and the entire Septentrion Environnement team. The SYMPHONIE model and the simulations produced are distributed by the national service SIROCCO (https://sirocco.obs-mip.fr) of CNRS-INSU coordinated by the ILICO research infrastructure. The simulations were performed using HPC resources from CALMIP (Grant P09115). CE is grateful to Pierre Chevaldonné and Thierry Perez for their constructive feedback.

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
