# Peer review of "Extreme sensitivity of the northeastern Gulf of Lion (western Mediterranean)"

_EGUsphere, 2024_

## Author Response (AR1)

**Reviewer 1**

Our warmest thanks to Francesco Pastor for his review and his very positive comments.

Answers to the questions are in red below and proposed text changes are in blue.

In this work, the authors examine the marine subsurface heatwaves recorded in the Gulf of Lion in summer 2022. The analysis is run from observations and, most importantly, numerical modelling. The authors describe the physical processes involved in a sound manner, properly separating different types of warm events depending on the wind conditions, and the presence of upwelling or downwelling. They also describe some biological impacts of the warm events in benthic ecosystems. This last part is the one that seems to me to be the least developed. From the title I would expect more information and discussion on the impacts, section 4.4 is too short. Maybe the authors should slightly change the paper title.

We propose to change the title from:

Extreme sensitivity of the northeastern Gulf of Lion (western Mediterranean) to subsurface heatwaves: Physical processes and devastating impacts on ecosystems in the summer of 2022

to:

Extreme sensitivity of the northeastern Gulf of Lion (western Mediterranean) to subsurface heatwaves: Physical processes and insights into effects on gorgonian populations in the summer of 2022

For all those reasons, I want to congratulate the authors for the work and recommend the publication of the paper with some very minor changes or clarifications (see below).

Minor comments

Línea 129-130 "but proved sufficient to achieve our objective." How was this proved? Did the authors run horizontal resolution sensitivity tests?

The sentence was unclear. We simply meant that the resolution provides a satisfactory representation of the measured temperature time series. We propose to change this sentence to:

The horizontal resolution in the area of interest is around 1900 m, which may seem coarse for an application very close to the coast, but has proved sufficient to represent rapid temperature variations, as will be shown in section 4.1 devoted to simulation evaluation (Figures 2 and 3).

Lines 237-238: Change 4.4.X to 4.3.X (section numbers seem to be misleading)

Thank you. Done

Line 230 "A sensitivity study will explore the temperature response to an extension of the wind period". What does this sentence mean? Does it refer to the next sections or to future work?

The sentence you refer to and the previous one were a declension of the last point quoted in the outline of section 4.3, namely the sensitivity of sub-surface warming to the duration of southeast winds (section 4.3.3). This was unnecessary and confusing. We have removed these two sentences. The justification for this test is given in section 4.3.3.

Lines 334-335: "During the summer of 2022, the atmospheric heatwaves that hit Western Europe gave rise to extreme marine heatwaves across the western Mediterranean". Is this supported by the authors work or from recent literature? I am not saying I disagree with this sentence but that it should be supported somehow. In the introduction, the authors cite the work of Guinaldo, if this is the base of the sentence it should be properly attributed.

Yes, you're right, the reference to Guinaldo in the introduction does correspond to this statement. We therefore propose to reintroduce it here as follows:

During the summer of 2022, the atmospheric heatwaves that hit Western Europe gave rise to extreme marine heatwaves across the western Mediterranean as shown by Guinaldo et al. (2023).

General remark for figures: I would change the font for the axis labels and scales to be clearer and more legible, but this is a matter of personal preference.
At least the first author agree! We changed the font of Figs 3, 4, 5 and 6.

**Reviewer 2**

We would like to warmly thank the reviewer for his/her comments and his/her patience in reading our text, which sometimes lacked clarity.
Answers to the questions are in red below and proposed text changes are in blue.

This work is an analysis of the atmospheric conditions and oceanic processes which define the extreme temperature conditions present in the summer of 2022 in the Mediterranean Sea. Using in-situ and model data to study regional-scale processes in the Gulf of Lion, the authors identify how the characteristics of the marine heatwave (duration, reoccurrence, depth and intensity) were shaped by local-scale processes (upwelling/downwelling and extension of currents). Data on the impacts on benthic communities is also presented.
In my opinion, the work is novel and of interest to those who study the Mediterranean Sea and marine heatwaves in general. The identified processes provide more nuance to the discussion of marine heatwave drivers, showing for example that winds can both "stop" or "redirect" heating. For this, I congratulate the authors. Although it is an interesting read, the writing makes it difficult to follow. Therefore, I suggest some minor revisions before publication.

**Specific comments**

The authors should improve the writing. While the main messages are there, some key points are not immediately clear, for example when describing temperature increases. Sections 4.3.1 and 4.3.2, in which the two processes are described, need to be made more explicit in my opinion. Please see below for some suggestions.

We've tried to simplify these two sections by removing unnecessary words and phrases. We hope that the text is now easier to understand. See also below for more details about the introduction of the sections.

The authors should provide more info on the ECMWF forecast model used.

It has been done in section 3. See the question and answer below.

Previous studies (see Darmaraki et al., 2024 and references therein) point to the crucial role of reduced latent heat anomalies, and to a lesser extent sensible heat, in the heat accumulation in the ocean (at least at the surface). Some data on latent heat during this event, if available at a suitable resolution, would be welcome, to either complement or challenge what has previously been reported.

The analysis of ocean-atmosphere individual fluxes is necessary and interesting to understand the processes involved in the various stages of the heatwave life cycle. In this way, Guinaldo et al (2023) have analyzed the 2022 heatwave at the large scale of the north-western Mediterranean Sea, using the ERA5 reanalysis. They showed that the total mixed layer temperature trend anomaly was due to anomalous turbulent heat fluxes (especially the latent one) explained by high specific humidity, hot air temperature and low wind speed.

In the coastal region studied in our article, the eastern Gulf of Lion, the mechanisms are very different. Warming is not local and is very rapid, as the warm water mass is simply advected from the east and eventually pushed down by the easterly wind and associated downwelling. We therefore believe that a 1D analysis of ocean-atmosphere fluxes is inadequate for our very localized study. To conclude, the latent heat flux has of course a remote effect to explain that the advected water mass if overheated in the surface layer. Locally in the studied region, it is of second order.

Moreover, Darmaraki et al., 2024 propose that intensified wind speeds are a driver of subsurface MHWs – the authors should discuss the importance of their findings in light of previous literature.

DARMARAKI, S., DENAXA, D., THEODOROU, I., LIVANOU, E., RIGATOU, D., RAITSOS, D., STAVRAKIDIS-ZACHOU, O.R.E.S.T.I.S., DIMARCHOPOULOU, D., BONINO, G., McADAM, R.O.N.A.N. and ORGANELLI, E., 2024. Marine Heatwaves in the Mediterranean Sea: A Literature Review. *Mediterranean Marine Science*, *25*(3), pp.586-620.

A sentence was added at the end of a small introduction to section 4.3.2 (see also the question about the writing difficult to follow):

Following Juza et al. (2022) and Darmaraki et al. (2024), such downwelling processes may indeed facilitate heatwaves vertical extension in the western Aegean Sea and the northeastern Crete.

**Technical corrections**

**Abstract**

Line 8: "an atmospheric situation" -> atmospheric conditions

Done

Line 9: Mediterranean Sea

Done

Line 10/11: "…down to depths of 30m which led to massive mortality…"

Done

**Intro**

Line 36: Full stop, not :

Done

**2 Main characteristics of the study site**

Line 83: low salinity

Done

Line 84: "originally" from the Atlantic?

Thank you. We propose to change the sentence:

The Gulf of Lion (Fig. 1) comprises a broad, crescent-shaped continental shelf enclosed between two straight coasts, the Ligurian coast to the northeast and the Catalan coast to the southwest, bordered by a narrow shelf and a steep continental slope, along which the Northern Current transports from northeast to southwest warm and low salinity waters from the south of the basin and originally from the Atlantic.

Line 105: Authors refer to a "A second type of wind" – this is confusing because two winds have already been discussed, and this "second type" is not shown on the map. Why not?

Yes it was confusing, as both Mistral and Tramontane are globally the first type of wind identified as northerly wind. We propose to change a little bit the sentence as follows:

After these northerly winds, the second type of wind blows from southwest to southeast…

And to add an arrow in Fig. 1 for this wind.

[Figure]

**3 Material and** Methods

Line 131: Why is the model driven by forecasts and not by analysis (if available)? How exactly are the forecasts used to drive the model (i.e. only lead time 0 is used)? Does this make the model a type of forecast itself or not?

The ECMWF forecasting model provides an analysis every 12 hours. From each of these analyses, forecasts are given every hour for several days. In order to remain close to the analyses (which are themselves close to reality) and to take into account the rapid variations in meteorological variables, we use the first 12 hours of forecasts after each analysis: the fields from 01.00 to 12.00, which follow the 00.00 analysis, and the fields from 13.00 to 24.00, which follow the 12.00 analysis.

We prefer this methodology to reanalysis (ERA5 for example), as the horizontal resolution is finer.

Yes, this protocol does indeed look like a forecast.

To  give more precise information, we changed the text as follows:

The model is initialized and forced in the Gulf of Cadiz (Atlantic Ocean) from daily operational oceanic analysis produced by MERCATOR OCEAN International and at the surface from the 12 hourly forecasts that follow the 00.00 and 12.00 analyses of the ECMWF operational meteorological model. COARE bulk formulas are used to compute the turbulent air/sea fluxes.

Line 138-193: No need to mention the mortality event again.

We agree. We removed it.

**4 Results**

Line 170: "model bias is largest/greatest"

Yes we changed by: the model bias is greatest at 5 m.

Line181: What does a "signature at 30m around +5°C" mean? An increase compared to what? The temperature increase in mid-August

seems closer to 10°C than 5°C.

You are right. We did consider only the final part of the event. We changed 5 by 8°C.

Line 195: "ECMWF model"

Done

Section 4.3: It is unclear which locations are being referred to here.

Yes, it's true. After Figure 3 which gives a larger geographic framework than Figure 2 corresponding to the Méjean site, section 4.3 returns to the Méjean site. We add the location in the first sentence:

The warmest event at the surface and at 20 m at Méjean...

Line 243: "temperature at 5m increased by 4 to 7°C" makes is sound like the final temperature reached was 7, which is confusing. Please reword.

We changed:

For each of these events, the temperature rise at 5 m ranged from 4 to 7 °C

Sections 4.3.1 and 4.3.2: The writing is a bit convoluted and difficult to follow. I would suggest (1) beginning with sentences which summarise the findings of the roles of upwelling/currents.

As said before, we've tried to simplify these two sections by removing unnecessary words and phrases. We hope that the text is now easier to understand.

As suggested by the reviewer, we also added an introduction to section 4.3.1 to explain the role of upwelling and associated current intrusions. In the first version, information on intrusions was given later in the text (the reference Barrier et al. (2016)) with a "suspense effect" that complicated the reader's understanding. We believe that this introduction now makes it possible to anticipate the processes at play. The new introduction is given below:

The upwellings that characterize the region studied here are mechanisms that a priori protect the coast from MHW by renewing surface water with deep cold water. However, Barrier et al (2016) have highlighted a collateral effect that occurs when the upwelling stops, in the form of intrusions of the Northern Current on the Gulf of Lion shelf with a delay of 1 day after the wind relaxation. The impact of these intrusions on the occurrence of heatwaves in the eastern Gulf of Lion is described in this section.

We also added an introduction to section 4.3.2:
Odic et al. (2022) showed a tight correlation between downwelling favorable winds and 35 m temperature anomalies along the Northwestern Mediterranean shorelines. They quantified that the 75th percentile of the WUDI index for downwelling corresponded to a alongshore wind of 5.2 m·s$^{-1}$ and a temperature response of +3°C at 35 m for stratified conditions. Here, we examine the impact of such downwelling on subsurface heatwaves. According to Juza et al. (2022) and Darmaraki et al. (2024), such downwelling processes may indeed facilitate the vertical extension of heat waves in the western Aegean and northeastern Crete.

Line 305: What was |well above average"? The downwelling?

The occurrence of southeasterly winds during summer 2022 is well above the mean summer occurrence. We propose to simplify:

As the occurrence of southeasterly winds is already high in August 2022, we limit ourselves to extending the southeasterly wind period by one day.

Line 313: "the period of strengthened winds"

We agree

Line 331: "populations were less affected"

Done

**5 Discussion and perspectives**

An interesting outcome of the study is that In the coastal environment, even nearby locations display very different MHW drivers and different responses to similar drivers – this could be highlighted.

Yes you are right. We already had this idea with this sentence:

On a regional scale (~ 200 km), we have shown considerable variability in surface and deep heatwaves for two reasons: (i) the channeling of northerly and easterly winds by the continental relief associated with the complex shape of the coastline, which produces localized upwelling and downwelling, and (ii) warming by the Northern Current, whose influence is great along the Ligurian coast and reduced in the western part of the Gulf of Lion.

and we propose to complete it with the blue text:
This high spatio-temporal variability of heatwaves in coastal areas contrasts with heatwaves in open seas, which are much more widespread and also more spatially homogeneous.

Line 359: "a numerical simulation"

Done

Line 370: "insights into"

Done

**Figures**

2: ECMWF operational model is not described anywhere – is this the same as the driver of the hydrodynamical model?

Yes it is the same. Please, note that the ECMWF forcing is a little bit more described now (see above). We also changed the figure legend:

Wind direction and intensity near Marseille from the ECMWF model (ocean model forcing) : e and f

Please add "Jun", "Jul", "Aug" etc to the x-axis labels, maybe adding vertical lines to denote the start of each month.

This was done. See new figure below

[Figure]

A1: What happened in summer 2014 to the data? I would change "data" to "in-situ".

A technical problem with the 40 m probe explains the absence of in-situ data. It is now specified in the legend.

We introduced at different places in-situ data including in Fig. A1